# Inferring thermospheric composition from ionogram profiles: A calibration with the TIMED spacecraft

Christopher J. Scott[1], Shannon Jones[1], Luke A. Barnard[1]

[1]Department of Meteorology, University of Reading, Reading, RG6 6BB, UK

*Correspondence to*: Christopher J. Scott (chris.scott@reading.ac.uk)

**Abstract.** We present a method for augmenting spacecraft measurements of thermospheric composition with quantitative estimates of daytime thermospheric composition below 200 km, inferred from ionospheric data, for which there is a global network of ground-based stations. Measurements of thermospheric composition via ground-based instrumentation are challenging to make and so details about this important region of the upper atmosphere are currently sparse. The visibility of

the F1 peak in ionospheric soundings from ground-based instrumentation is a sensitive function of thermospheric composition. The ionospheric profile in the transition region between F1 and F2 peaks can be expressed by the 'G' factor, a function of ion production rate and loss rates via ion-atom interchange reactions and dissociative recombination of molecular ions. This in turn can be expressed as the square of the ratio of ions lost via these processes. We compare estimates of the G factor obtained from ionograms recorded at Kwajalein ($9^o$N, $167.2^o$ E) for 25 times during which the TIMED spacecraft

recorded approximately co-located measurements of the neutral thermosphere. We find a linear relationship between $\sqrt{G}$ and the molecular: atomic composition ratio, with a gradient of 2.55 ±0.40. Alternatively, using hmF1 values obtained by ionogram inversion, this gradient was found to be 4.75 ± 0.4. Further, accounting for equal ionisation in molecular and atomic species yielded a gradient of 4.20 ± 0.8. This relationship has potential for using ground-based ionospheric measurements to infer quantitative variations in the composition of the neutral thermosphere via a relatively simple model.

This has applications in understanding long-term change and the efficacy of the upper atmosphere on satellite drag.

## 1 Introduction

A small fraction of the Earth's upper atmosphere, the thermosphere, is ionised, principally by solar extreme ultra violet and x-ray radiation, to form the ionosphere. Continuous measurements of the Earth's ionosphere have been made since the early 1930s (e.g. Gardiner et at, 1982) exploiting the fact that ionisation reflects high frequency (HF) radio waves. In this way,

detailed records have been obtained of the long-term variation of the ionosphere in response to changes in season, solar activity, space weather events and phenomena such as solar eclipses. Initiatives such as the International Geophyscial Year in 1957 enabled a comprehensive global coverage of routine ionospheric measurements that, despite some decline in the number of observing stations, continues to date.

While the ionosphere makes up only a small fraction (~0.001%) of the upper atmosphere, measuring the neutral thermosphere is more challenging. Ground-based measurements of the thermosphere have been made via optical detection of atmospheric airglow from which temperature and winds can be inferred (e.g. Burnside et al, 1982; Meriwether et al, 1983; Griffin et al, 2008). Though too high for conventional weather balloons, in-situ measurements of thermospheric composition, temperature and winds have been made using sub-orbital rockets, usually conducted on a campaign basis, which provide

only a highly localised snapshot of thermospheric conditions (e.g. Spencer and Carignan, 1988). More recently earth-orbiting spacecraft such as the Thermosphere Ionosphere Mesosphere Energetics and Dynamics (TIMED) spacecraft (Kusnierkiewicz, 2003) have made measurements of thermospheric composition, temperature and winds from orbit. Far-ultraviolet remote sensing provides information on the integrated column $O/N_2$ ratio or height profiles of O and $N_2$ concentrations (via observations of the airglow profile on the limb of the Earth). Such measurements can build up a global

picture of the thermosphere with maps composed from measurements made over many orbits. In 2018 the Global Observations of the Limb and Disk (GOLD) instrument (Eastes et al, 2017), hosted by the STS-14 commercial spacecraft, was launched into a geostationary orbit from where it makes column-integrated measurements of the thermosphere over an entire hemisphere and height profiles at the limb. Despite these advances, information about thermospheric composition is limited to dayside above around 200 km. This paper proposes a method of augmenting these spacecraft measurements with

estimates of thermospheric composition below 200 km via a global network of ground-based ionospheric observatories.

The thermosphere directly impacts modern technology such as via frictional drag on spacecraft and, through its influence on the ionosphere, radio communications and the accuracy of Global Navigation Satellite Systems (GNSS). Thermospheric composition measurements are also needed to understand the seasonal variation of the ionosphere (e.g. Rishbeth and Setty

1961; King, 1961b; Rishbeth and Kervin, 1968) which differ with geographic location and have been shown to exhibit long-term changes (Bremer, 2004; Scott, Stamper and Rishbeth, 2014; Scott and Stamper, 2015).

Given the potential applications for measurements of the neutral thermosphere, and the influence this has on the ionosphere,

it is desirable to investigate whether the ionospheric measurements can be used to measure the thermosphere by proxy. Recent studies (Mikhailov et al, 2012; Mikhailov and Peronne, 2016; Peronne and Mikhailov, 2018) have used a sophisticated model containing comprehensive ion chemistry to generate fits to ionospheric profiles. They have shown good agreement between the neutral density derived from their model and the thermospheric density as measured by the CHAMP spacecraft (Bruinsma, S., et al, 2004).

In this paper we investigate the potential of a more simplified technique developed in the 1960s (King, 1961; King and Lawden, 1964; Rishbeth and Kervin, 1968, King, 1969) in which the shape of the ionospheric profile measured by ground-based instrumentation is used to infer relative changes in the thermospheric composition at the height of the ionospheric F1 layer peak. A comparison with co-located measurements of the thermospheric composition from the TIMED spacecraft

provides an opportunity to determine if the shape of the ionogram profile in the F1 to F2 layer transition can be used to infer a quantitative estimate of the thermospheric composition at the same altitudes.

## 2 Theoretical background

Ions and electrons within the ionosphere exist in a state of dynamic equilibrium, with ions being produced by photoionisation and lost through recombination. Which particular loss process dominates depends on the composition of the background thermosphere. The atmosphere above the turbopause at around 100km becomes sufficiently tenuous that the gasses are no longer mixed through collisions and each gas species diffuses into a hydrostatic equilibrium according to its molecular or atomic mass. Heavier molecules such as $N_2$ and $O_2$ dominate at lower thermospheric altitudes (around 100km) while atomic oxygen, O, becomes the dominant neutral species at greater altitudes (around 300km).

Early work on the ionospheric continuity equation (Rishbeth and Garriott, 1969, and references therein) for the concentration of electrons, N, atomic ions $N_{A+}$ and molecular ions $N_{M+}$ assumed that molecular ions in the lower thermosphere were lost through dissociative recombination while the ion-atom interchange reaction occurring at F2 altitudes where $O^+$ ions dominate, involved interaction with a molecular gas, of concentration n[M]. This process dominates in the upper thermosphere since molecular ions produced by photo-ionisation are rapidly removed by dissociative recombination. Given these assumptions, it was possible to write the continuity equations for N, $N_{A+}$ and $N_{M+}$ as;

$$\frac{dN}{dt} = q - \propto NN_{M+} \tag{1}$$

$$\frac{dN_{A+}}{dt} = q - \gamma n[M]N_{A+} \tag{2}$$

$$\frac{dN_{M+}}{dt} = \gamma n[M]N_{A+} - \alpha NN_{M+} \tag{3}$$

with charge neutrality requiring that;

$$N = N_{A+} + N_{M+} \tag{4}$$

where q is the ion production rate, α is the 'square law' loss coefficient associated with dissociative recombination and γn[M] is the loss coefficient for atomic ions through ion-atom interchange. Assuming equilibrium conditions (d/dt=0) and writing β=γn[M], the ratio of atomic and molecular ion concentrations can then be written;

$$\frac{N_{A+}}{N_{M+}} = \alpha N/\beta \tag{5}$$

Using equation (5) to substitute for $N_m{}^+$ in equation (1) and similarly substituting for $N_A{}^+$ from equation (2), a quadratic
expression for the electron concentration can be obtained.

$$\alpha\beta N^2 - \alpha q N - \beta q = 0 \tag{6}$$

The positive root of this equation gives the expression;


$$N = \left(\frac{q}{2B}\right)\left[1 + \left(1 + \frac{4\beta^2}{\alpha q}\right)^{\frac{1}{2}}\right] \tag{7}$$

This reduces to

$$N = N_\alpha = \left(\frac{q}{\alpha}\right)^{\frac{1}{2}} \; if \; 4\beta^2 \gg \alpha q \tag{8}$$
and

$$N = N_\beta = \frac{q}{\beta} \; if \; 4\beta^2 \ll \alpha q \tag{9}$$

Ionospheric layers embedded within the thermosphere therefore undergo different loss processes. At E-region altitudes, around 100 km, the dominant neutral species are molecular, and so equation (8) dominates which results in rapid recombination ($\sim$ seconds) for a given molecular ion ($O_2{}^+$ and $NO^+$). This reaction is fast because it is not restricted to a specific relative energy between the ions and electrons since any additional energy breaks the molecular bonds, generating atomic neutrals. This dissociative recombination of ionisation associated with molecular ions is rapid compared with the loss
at higher F-region altitudes (around 300km) where neutral atomic oxygen dominates. Recombination with atomic oxygen either requires a third body in the collision to absorb any excess kinetic energy or that the electron and ion have specific energies that enable an electron to be captured by the ion. Consequently, such reactions are much slower than loss of ionisation through dissociative recombination. Atomic ions can also undergo charge-exchange reactions with molecular neutrals, with the resulting molecular ions rapidly recombining. As a consequence, equation (9) applies with atomic oxygen
ions tending to dominate in the upper ionosphere under equilibrium conditions.

Given these two regimes, there must be an altitude at which the equilibrium reaction transitions between the domains of the $\alpha N^2$ and the $\beta N$ loss processes. This occurs in the lower F region, where the peak level of F-region ion production occurs. Ratcliffe (1956) demonstrated that this could account for the splitting of the F layer into the F1 and F2 components.


From equation (7) it can be shown that the parameter $\beta^2/\alpha q$ determines the shape of the height profile in electron concentration. Defining $G$ as the value of this parameter at the level of peak production, and using equations (8) and (9) it is possible to show that;

$$G = \frac{\beta^2}{\alpha q} = N_\alpha^2/N_\beta^2 \qquad (10)$$

If K is defined as the ratio of scale heights of the ionisable gas and the linear loss coefficient, $\beta$, K depends on the molecular gas involved in the ion-atom interchange ($O_2$ or $N_2$). Since the dominant ionisable gas is atomic oxygen, K=28/16 =1.75 for $N_2$ and 2 for $O_2$ if $\beta$ is assumed to be independent of temperature.


If G and K are constant, changes in q and $\beta$ can affect the magnitude and vertical position of the electron concentration height profile, but do not affect its shape.

Figure 1 (after figure 28, Rishbeth & Gariott, 1969) presents the vertical profile of electron concentration for a range of 
values of G. For low values of G the transition is smooth, with a large ledge appearing for G=4 which becomes more pronounced for larger values of G.

King (1969) went further and considered the fraction of production, F, occurring in the atomic part of the atmosphere, resulting in a modified form of equation 6;

$$\alpha\beta N^2 - \alpha F q N - \beta q = 0 \qquad (11)$$
By also considering the ratio of production rates in the molecular and atomic fractions of the ionosphere, R, at a given level, King (1969) was able to extend this simple model to account for this, resulting in a revised definition of G;

$$G_K = (1 + R)^2 \frac{\beta^2}{\alpha q} \qquad (12)$$

He concluded that while the inclusion of these terms would affect the overall quantitative estimate of the thermospheric molecular:atomic ratio, it did not significantly affect the shape of the resulting modelled ionosonde profiles. King also suggested that a parameter equivalent to $\sqrt{G}.q_o/\alpha_o$ ( here referred to as $G^*$) should be related to molecular density, where

$$G^* = (1 + R) \frac{\beta_o}{\alpha_o} \qquad (13)$$

Between the F1 and F2 layers a transition occurs from a thermosphere that is predominantly molecular to one where neutral atomic oxygen dominates, with the dominant loss process changing from the square law loss coefficient α to the linear loss coefficient β. The exact nature of this transition has a marked impact on the visibility of the F1 peak observed in vertical profiles of ionospheric electron concentration as measured by ionospheric sounders, with such a ledge resulting in a prominent 'cusp' for G > 1.

While the general behaviour of the F1-F2 transition is therefore controlled by thermospheric composition, transient features generated by atmospheric dynamics, such as Travelling Ionospheric Disturbances (TIDs), have long been known to temporarily affect electron concentration gradients within the ionospheric profile (e.g. Munro, 1950, Rawer, 1959). While caution should therefore be used when interpreting the shape of any individual ionospheric profile, the transient nature of a TID is revealed by viewing an ionogram in the context of the diurnal time series. In this way, the impact of TIDs can be minimised in any investigation for which a sufficiently large data set is available, though their presence would add to the scatter into any quantitative relationship between estimates of the G parameter and thermospheric composition.

**Rishbeth and Kervin (1968) conducted a qualitative assessment of changes to the G parameter for three stations; Anchorage, Washington and Grand Bahama. They observed a seasonal change in G which followed the expected behaviour in ionospheric production rate, q, when corrected for solar zenith angle and sunspot number. They concluded that β was the most likely factor to be causing the observed changes which in turn suggested a variation in thermospheric composition or temperature. Changes in thermospheric composition are also consistent with observations of the F-region seasonal anomaly (Rishbeth and Seti, 1961; King 1961b; King, 1970).3 Ground-based ionospheric monitoring- the ionosonde**

Since the early 1930s, routine observations of the distribution of ionisation with height in the Earth's atmosphere have been undertaken using ionospheric sounders known as ionosondes (Gardiner et al, 1982). These ionosondes rely on the fact that a radio signal transmitted vertically will be returned from an altitude at which the local plasma concentration matches the frequency of the transmitted radio pulse. The electron concentration, $N$, (m$^{-3}$) is related to the radio frequency, $f$, (Hz) by the relation;

$$f = 8.98\sqrt{N} \tag{11}$$

Typical electron concentrations within the Earth's ionosphere correspond to radio frequencies in the high frequency (HF) waveband ~ 0.5 – 20 MHz. By transmitting a range of radio frequencies and assuming that the radio pulses are travelling at the speed of light in a vacuum, a height profile of the electron concentration in the ionosphere can be estimated from the time of flight versus radio frequency. Such a plot is referred to as an ionogram. Since the radio pulses are not travelling through a vacuum but a weakly ionised medium, their time-of-flight is extended through interaction with the local plasma. The heights on an ionogram are therefore referred to as virtual heights, h', which are greater than the equivalent true heights, h, depending on the amount of ionisation through which the pulse has had to travel. As the transmitted radio frequency

approaches the resonant plasma frequency at the peak of an ionospheric layer it is further delayed by its resonance (frequent absorption and re-emission by the local electrons) and the time of flight becomes effectively infinite. This leads to distinct cusps on an ionogram at the peak frequency of each layer (see example in figure 2). Such critical frequencies are denoted foE, foF1, foF2 for the E, F1 and F2 layers respectively and have long been used to represent the peak frequency returned by the layer, and via equation 11, the peak electron concentration. As detailed above, the visibility of the foF1 cusp in ionograms is controlled by the *G* parameter, which in turn is determined by the thermospheric composition in the transition region between the F1 and F2 layers.

## 3.1 Estimating values of G from an ionogram

Analysis of the F1-F2 transition region was instigated by King (1961), in which the ionospheric electron concentration profile was compared with a set of theoretical curves derived for a range of values of G. This work demonstrated that it was possible to determine relative changes in G throughout the day, from which changes in thermospheric composition could be inferred. This technique was further elaborated (King and Lawden, 1964) to one in which templates were produced and matched with ionospheric profiles. The curves on these templates were created assuming a constant scale-height, $H_v$, of 30 km at the bottom of the transition region and a vertical scale height gradient, $\Gamma$, that varied with reduced height, $Z$, as $H_O = H_v exp(-\Gamma Z_v)$. These templates, produced for a range of values of $\Gamma$ and $G$, were compared with each ionospheric profile and the template that best matched the curve was used to identify the associated values of $\Gamma$ and $G$. Rishbeth and Kervin (1968) used a similar technique to investigate the diurnal and seasonal variation of G from which they inferred changes to the neutral thermospheric composition or temperature.

Lawden (1969) subsequently investigated problems associated with the assumptions made in producing the templates, namely that the ionisation was assumed to be entirely $O^+$, that the ionising radiation was monochromatic and that the loss rate, $\beta$, was inversely proportional to the temperature. He concluded that the relative production rates of $N_2^+$ and $O^+$ was very important, as was any temperature variation in the reaction rates. King (1969) addressed the validity of these assumptions and concluded that the assumptions were indeed adequate for expressing the shape of the F1 region quantitatively.

The purpose of this paper is to apply the methods of King and Lawden (1968), as used by Rishbeth and Kervin (1968) and King (1969) to modern ionogram traces so that estimates of G, $G_K$ and $G^*$ can be made. Instead of producing templates however, a best-fit curve was found by minimising the residuals between the data and model profiles within the transition region of each ionogram. This method is largely the same as used previously except that neither the initial scale height nor the foE/foF1 ratio were fixed.

The production and loss functions in the ionospheric profile are;

$$q = q_o \exp\left[(1 + \Gamma)(1 - z - e^{-Z})\right] \tag{12}$$

$$L = \beta n / \left(1 + \frac{\beta}{\alpha N}\right) \tag{13}$$

where

$$\beta = \beta_o \exp\left[-(k + \Gamma)z\right] \tag{14}$$

Assuming equilibrium, q = L, these equations can be used to form a quadratic expression in N√(α/q) (King and Lawden, 1968);

$$\left(N\sqrt{\alpha/q}\right)^2 - \left(N\sqrt{\alpha/q}\right)\sqrt{q\alpha/\beta^2} - 1 = 0 \tag{15}$$

Alternatively, when adopting the method of King (1969) in which a fraction, F, occurs in the atomic part of the atmosphere, this expression becomes;

$$\left(N\sqrt{\alpha/q}\right)^2 - \left(NF(\alpha/\beta)\sqrt{q/\alpha}\right) - 1 = 0 \tag{16}$$

Here we adopt the practice of Rishbeth and Kervin (1968), and define $H_1$ as the scale height of atomic oxygen at the peak of ion production for the layer, but rather than estimating the scale height profile assuming a constant vertical gradient, $\Gamma=dH/dh$, we instead adopt the practice of King and Lawden (1964) in which the scale height gradient varies exponentially with reduced height although we take the reference scale height, $H_1$, at the peak of the F1 layer. Unlike these authors, we

include $H_1$ as a variable during our fitting process. We also assume that $N_2$ is the dominant molecular neutral species so that K=1.75. Since we have no prior information about the temperature profile, we adopt the assumption of these earlier analyses that the loss rate, α, is assumed to vary inversely with temperature and therefore H.

In order to produce templates, Rishbeth and Kervin (1968) also assumed a fixed ratio between foE and foF1 (by inspection

of a long time series of data from the station under study). This was to enable a reasonable estimate of the curve at the lower end of the transition region. Since we are carrying out a best-fit, we allow this to be a variable in our analysis.

Having estimated the ionospheric profile in the transition region, we use the critical frequency of the E-region to produce a simple model of the E-region represented as a 10km thick slab of ionisation at the height of the E-layer (assuming

h'E≈hmE). Since we have no information about the ionisation within the E-F valley, we initially assume a true height of 200 km for the F1 layer and include a variable offset, *h_offset*, which is used to best match the difference in virtual height between the E and F1 layers.

Having created an ionospheric height profile by whichever method, this then needs to be converted to virtual height for comparison with each ionogram. The virtual height, h', and the true height, h, are related by the integral;

$$h'(f) = \int_0^f \mu'(dh/df_N)df_N + h(0) \tag{15}$$

Where h(0) is the height at the base of the ionosphere below which it is assumed that $f_N$=0 and $\mu'$ is the group refractive index, a complex function of plasma and gyro frequencies derived from the Appleton Hartree equation (e.g. Storey, 1959; Rishbeth and Garriott, 1969). This process therefore requires an estimate of the magnetic field, B, for which the International Geomagnetic Reference Field (IGRF, Thébault et al, 2015) was used for the time and location of each measurement.

## 3.2 Analysis of the ionogram data

To compare the model ionogram with real data, first some parameters (foE, h'E and the frequencies corresponding to h'F1 and h'F2) need to be scaled from the ionogram data and used as input to the ionogram model. These parameters remain fixed throughout the fitting process. Having created a model ionogram using default initial input parameters ($\Gamma_1$=0, G,$G_K$ or G*=1, $H_1$=300 km, foF1=foE/0.7 and h_offset = 0), the model and data-derived ionogram traces are compared and the differences between the two curves within the transition region (between two frequencies denoted by the lowest virtual height of the F1 and F2 layers) was used to calculate a mean square residual. The input parameters are then iterated to minimise this residual and the resulting 'best-fit' parameters stored. Despite there being five input parameters, each affects the profile in different ways. The main influence of varying $\Gamma_1$ is to affect the difference in virtual height between the F1 and F2 layers. Varying the G parameter, as expected, alters the visibility of the F1-F2 transition, $H_1$ affects the fit to the F1 profile and h_offset adjusts the height between the E and F1 layers. Having ascertained the input parameters that resulted in the best fit to the data, uncertainties in G and $H_1$ were estimated by iterating these values until the mean square residual was doubled. The four parameters are not completely independent however and there are many local minima within the fitting surface. Automated minimisation techniques tended to find a local minimum closest to the initial model parameters and so for the purposes of this demonstration, the minimum residual was determined by manual inspection and iteration. This manual check ensured that the minimum difference between model and data ionograms best represented the shape of the curve around the F1-F2 transition. While it is highly desirable to automate this part of the process, tests showed that independent manual fitting of the same ionograms obtained results that were within the quoted uncertainties.

## 4 Thermospheric Profiles from the TIMED Spacecraft

Data from the Global Ultra-Violet Imager (GUVI, Yee et al, 2003) on board the Thermosphere Ionosphere Mesosphere Energetics and Dynamics (TIMED) spacecraft (Kusnierkiewicz, 2003), were used for comparison with the ground-based

ionospheric parameters derived in this study. This mission, launched in December 2007 was designed to study the ionospheric and neutral atmosphere interactions in the mesosphere and lower thermosphere. The TIMED spacecraft carried four remote-sensing instruments, in a circular Earth orbit at an altitude of 625 km with an inclination of $74.1^{o}$. The four instruments are; GUVI, a cross-track scanner measuring spatial and temporal variations of temperature and thermospheric gas concentrations in the lower thermosphere; The Solar Extreme ultraviolet Experiment (SEE) which measures solar radiation; The Sounding of the Atmosphere using Broadband Emission Radiometry (SABER), measuring infra-red emissions below 120 km and the TIMED Doppler Imager (TIDI) measuring wind direction and speed.

## 4.1 The Global Ultra-Violet Imager

The GUVI instrument is a scanning spectrograph sensitive to emissions in the far ultraviolet region (115 -180 nm) of the electromagnetic spectrum. It was designed to detect airglow emissions in the Earth's upper atmosphere. Within this region of the spectrum there are strong emissions from atomic hydrogen, (H) at 121.6 nm; atomic oxygen (O) at 130.4 and 135.6 nm and molecular nitrogen ($N_2$) at 165 and 185 nm. Absorption by molecular oxygen ($O_2$) occurs across the spectrum and this must be accounted for in determining O and $N_2$ concentrations (Meier et al, 2015). The GUVI instrument has a field of view of $11.78^{o}$ and used a mirror to sweep this through an arc up to $140^{o}$ perpendicular to the spacecraft orbit with a spatial resolution of around 8km. After 10 million such scans, the mirror failed in 2007. The instrument continues to operate in a fixed look direction, at about $47^{o}$ from nadir. When scanning was possible, three regions were typically monitored during each orbit; the daytime low-to-mid latitude thermosphere, the night time low-to-mid latitude thermosphere and the high latitude auroral zone.

Of interest for this study are thermospheric composition and temperature profiles derived from limb observations using GUVI data (Meier et al, 2015). Due to the nature of the orbit and varying solar activity, these measurements are made under a range of solar illumination and solar and geomagnetic conditions. In order to estimate altitude profiles of the individual gas species, careful calibration is required before the nonlinear relationships between emission rates and number densities can be deconvolved. There is insufficient information to employ tomographic inversion techniques, and so additional information from the NRLMSISE-00 model was used to constrain the extraction of geophysical parameters from the column emission rates. The technique is robust for measurements made at locations with latitudes below $60^{o}$ (to avoid auroral contamination) and with a solar zenith angle less than $80^{o}$. Data from the epoch 2002-2007 have been analysed in this way and are made available at http://guvitimed.jhuapl.edu/data_products. These data products are subject to ongoing updates as resources allow. Each file contains data detailing the vertical profiles of thermospheric temperature and the concentrations of the neutral constituents $N_2$, $O_2$ and O below 310 km. While each profile represents a line-of sight view through the airglow emission profile, it is assumed that there is no horizontal variation (an assumption that is most appropriate for lower latitudes) and the location of a given profile is defined as the tangent point at the peak of the airglow layer, around 160km.

While the location of the spacecraft at any given time can be determined from the online data archives, the location assigned to each of the limb observations is stored within each file. At the time of writing, this required each data file to be examined in order to determine the location and timing of such measurements. In order to enable rapid comparison with ground-based measurements, a database was created containing the geographic location of all limb scans within the dataset. It was subsequently possible to determine the times at which measurements were being made in the vicinity of ground-based ionospheric monitoring stations for which data were available via the Global Ionospheric Radio Observatory (GIRO; Reinisch and Galkin, 2011). While the coincidence of observations is a function of the timing of both spacecraft and ionospheric observations, the nature of the spacecraft orbit meant that there were far more overpasses for low-latitude ionospheric monitoring stations than for those at high latitudes. Since the purpose of this study is to look at any potential correlation between satellite measurements of thermospheric densities and the shape of ground-based ionospheric profiles, it is advantageous to use measurements far from the auroral zones and their associated geomagnetic activity. Under more auroral conditions, the simple assumptions made in this study may not hold and the enhancement of gravity waves would likely distort the ionospheric profile. In addition, it appears that the observations are often made at high zenith angles at these latitudes, making them less useful both for interpreting the TIMED measurements but also in assuming dynamic equilibrium within the ionosphere. That is not to say that low-latitude ground-based ionospheric observations are without their own complications. Additional stratification, the F1.5 layer, can occur in this region (e.g. Yadav et al, 2012) while sporadic E and spread-F are also commonplace. Despite these limitations, the station for which there were most coincident observations was at Kwajelein ($9^o$ N, $167^o$E) situated within the Marshall Islands in the Pacific. Here there were 59 overpasses in total for which there were ground-based ionospheric data within 15 minutes and co-located within $5^o$ latitude and longitude. Five of the ionospheric profiles were obscured by sporadic E (Es) while clear ionospheric profiles were apparent in the remaining 54. In order to increase the likelihood that the ionosphere was in a state of dynamic equilibrium, the data were further restricted to those for which the solar zenith angle was less than $60^o$. The remaining 29 ionograms provided the best possible data set with which to compare ground-based observations with thermospheric profiles from the TIMED spacecraft while preserving a data set that was sufficiently large to ensure the comparison was meaningful.

## 5 Comparison between spacecraft and ionospheric data

In order to minimise the introduction of unconscious bias into the analysis, the height profile from each of the Kwajalein ionograms was manually scaled and the best-fit profile of the transition region was determined (through iteration of the model ionograms as detailed in section 3.2), without reference to TIMED data. Having fitted each of the ionograms, the scale height at the F1 peak determined from the fit was used to estimate the thermospheric temperature. This temperature was in turn used to identify the altitude within the TIMED data from which the molecular to atomic concentration ratio was estimated. For the purposes of this analysis, this ratio was taken to be $[O]/([O_2]+[N_2])$. Any fit that resulted in an altitude of less than 125 km for the height of the F1 peak was deemed unphysical and removed from the analysis. This was the case for 4 of the traces which corresponded to ionograms for which a sporadic E layer or bifurcation of the E layer was seen.

As β is influenced by changes to thermospheric composition, a plot of √G versus the molecular to atomic ratio should be a linear relationship through the origin. Such a plot is presented in figure 3 for the 25 cases used in this study. While there is some scatter, for the combination of reasons set out above, the data do indeed follow a linear trend, with a gradient of 2.55 ±0.4. This, and all subsequent gradients were determined using Theil Sen fitting, which accounts for outliers (Theil 1950;

Sen, 1958). Uncertainties in the gradients were determined by conducting 1000 runs in which values were randomly sampled from the uncertainty distributions in two dimensions. All lines were assumed to run through the origin.

The assumption of 100% $O^+$ ions used to obtain the above relationship is consistent with the original analyses of King and Lawden (1964) and Rishbeth and Kervin (1968). However, as the F1 peak represents the transition between square and linear

loss terms, it is likely that there will be a significant fraction of both molecular and atomic ions. If information about the true height of the F1 peak is available, this can be used instead to determine the height at which comparison with the satellite data should be made. The value of ion temperature at this height retrieved from the spacecraft data can then be used to determine the average ion mass from the scale height fitted to the F1 peak. Assuming a value of 31 a.m.u for the molecular ions (an average between $NO^+$ and $O_2^+$ ions) enables estimation of an additional parameter; the fraction of atomic ions. There are two

potential estimates of hmF1; self-consistently from the fit to the ionospheric profile or independently via the profile inversion algorithm ARTIST. While there is a reassuring degree of correlation between the two estimates, the fitted values are ~5% lower than those produced by ARTIST. The ARTIST true heights likely better represent reality than those from the fit due to the simplistic nature of the E-layer reconstruction and valley approximations used to generate the electron concentration profiles in the model. Repeating the comparison with TIMED data using the ARTIST estimates of the F1 layer

height generates the relationship presented in figure 4. The gradient for the best fit to these data is 4.75 ± 0.8 . The fraction of $O^+$ ions estimated at hmF1 (for those 20 profiles where ARTIST was able to estimate) produced realistic values for this quantity. While a direct comparison with in-situ measurement or independent instrumentation such as Incoherent Scatter Radar is needed to validate these individual estimates, a comparison with ion composition obtained from the International Reference Ionosphere model for these dates yields a similar distribution at the altitude of the F1 peak.

Having determined these quantitative comparisons of neutral composition between ground-based and satellite measurements using the simple model presented by Rishbeth and Kervin (1968) we extend our analysis further to consider equal production in both molecular and atomic species. For this the fraction, F, was set to 0.5 and the ratio, R, was set to 1. Figure 5 presents the relationship between $√G_K$ and the thermospheric molecular to atomic ratio. The distribution of the points is similar to that seen for the more simple analysis (figure 4), with the gradient modified slightly to 4.2 ± 0.8. Further, the relationship of the

alternative parameter G* with the thermospheric molecular to atomic ratio was also investigated. These results are presented in figure 6, with a best-fit gradient of 0.86 ± 0.14.

## 6 Summary and discussion

It is apparent that this simple analysis has adopted many of the assumptions used during the analysis of King and Lawden (1964) and Rishbeth and Kervin (1968). From a modern perspective, more is now known about the temperature dependence of the loss rates pertinent to this study. Inclusion of this information, along with assumptions made about the rate of change of scale height with altitude, was prevented by a lack of independent temperature data. Were such information available (for example from co-located ground-based optical measurements of atmospheric airglow) these assumptions could be addressed. Similarly the assumption that the ionospheric profile within the transition region is dominated by atomic ions will only be applicable to mid to low latitude stations under conditions of low geomagnetic activity. Estimating the height of the F1 layer independently using ionogram inversion or from the model fit itself produces a consistent relationship between the G parameter and the molecular:atomic ratio. The presence of gravity waves and bifurcation of layers in ionograms can influence the shape of the trace and the visibility of the G parameter. These add scatter to the points plotted in figure 3. In particular, if there are additional layers apparent in the ionogram between the E and F1 peaks, this can lead to an underestimate of the scale height at the F1 peak, leading either to an underestimate of the local temperature and therefore the height of the F1 peak or artificially raise the estimated fraction of $O^+$ ions, depending on which method is applied.

These techniques were developed before comprehensive in-situ measurements of the thermosphere and ionosphere had been made. Indeed many of the reaction rates involved were only known approximately at the time (King, 1961; King and Lawden, 1964; Rishbeth and Kervin, 1968). Modern measurements of these reaction rates reveal that those involving dissociative recombination of molecular ions are dependent on the electron temperature (e.g. Schunk and Nagy, 2000 and references therein). With no information concerning the neutral, ion or electron temperatures from the ionosonde data, the variation of these with height is not known, though at lower ionospheric altitudes under daytime quiet geomagnetic conditions the ion and neutral temperatures can be considered as being in equilibrium, with the temperature of the photo-electrons being elevated above these. The simple model employed in the current analysis assumes the reaction rates will simply vary inversely with the scale height, which is a function of ion temperature and inversely proportional to the ion mass. Since the ion mass decreases with height, the scale height will vary more rapidly with height than through variations in ion temperature alone, and this may be sufficient to emulate the variation in height of the loss coefficient, α. In order to investigate this, the International Reference Ionosphere (IRI; e.g. Bilitza, 2018)  was used to estimate the ionospheric conditions above Kwajalein for the dates used in this study. From these runs, height profiles of mean molecular loss rate were calculated using electron temperatures and ion compositions estimated from the IRI. These profiles are presented as solid lines in figure 7, along with those estimated using the scale height approximation. While there is some scatter about the IRI estimates, there is no systematic offset between the two, suggesting that this approximation will contribute to the scatter of points in the comparison between G and thermospheric composition. Furthermore, there is no correlation between fitted

values of G and the scale height gradient, Γ, obtained from the model, suggesting that any uncertainty in one parameter will not lead to a systematic uncertainty in the other.

When extending the model to include equal production in both atomic and molecular species, the resulting relationship with the thermospheric composition was modified slightly (from $4.75 \pm 0.8$ to $4.20 \pm 0.8$) both are consistent within the uncertainty estimates. This is consistent with the conclusion of King (1969) that the inclusion of production in both atomic and molecular species did not significantly affect the values of G estimated from the ionospheric profiles.

Despite these simplifications, we have shown that there is an empirical relationship between various definitions of the G parameter estimated from ionograms and the thermospheric molecular:atomic concentration obtained from satellite data. This result requires further validation against modern ground-based instrumentation such as Incoherent Scatter Radar (for which similar studies have been conducted; Oliver, 1979) and, while there is much to do to advance the technique itself, it suggests that this relatively simple approach using ground-based ionospheric data could be used to infer quantitative information about the neutral thermosphere. With the recent launch of the GOLD instrument (Eastes et al, 2017) there is further potential for detailed comparison of spacecraft measurements of the thermosphere with ground-based measurements of the ionosphere. Calibrating these ground-based stations would enable investigation into the spatial scale of thermospheric variability as well as a quantitative method of studying long-term changes in thermospheric composition. Until now, such studies have been limited to qualitative comparisons (e.g. Rishbeth and Kervin, 1968, Scott, Stamper and Rishbeth, 2014, Scott and Stamper, 2015).

**Data availability**

The ionospheric data used in this analysis were obtained from the Digital Ionogram Database (https://ulcar.uml.edu/DIDBase/). Data from the TIMED spacecraft were obtained via http://guvitimed.jhuapl.edu/data_products.

**Author contribution**

CJS Conceived the project, developed the code, conducted the analysis and wrote the draft manuscript, LAB contributed to the data acquisition and analysis software and SJ conducted data retrieval and analysis. All authors contributed to the final manuscript.

**Competing interests**

The authors declare that they have no conflict of interest.

## Acknowledgements

The data used in this study were obtained from the Global Ionospheric Radio Observatory (GIRO: http://spase.info/SMWG/Observatory/GIRO). The Kwajalein ionosonde station data is managed by NOAA. The GUVI data used here are provided through support from the NASA MO&DA program. The GUVI instrument was designed and built by The Aerospace Corporation and The Johns Hopkins University. The Principal Investigator is Dr. Andrew B. Christensen and the Chief Scientist and co-PI is Dr. Larry J. Paxton whom we thank for their support and encouragement. CJS would like to dedicate this work to Henry Rishbeth who was an enthusiastic supporter of the UK ionospheric monitoring programme. Henry provided much insight into this work through informal conversations long after he had formally retired. This work ultimately resulted from the careful scrutiny with which Rita Blake MBE, the data scaler for the Slough/Chilton data set, examined the data. Her familiarity with the data and awareness that scientific progress results from the things which could not be explained by the current published literature has led to much scientific progress.

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

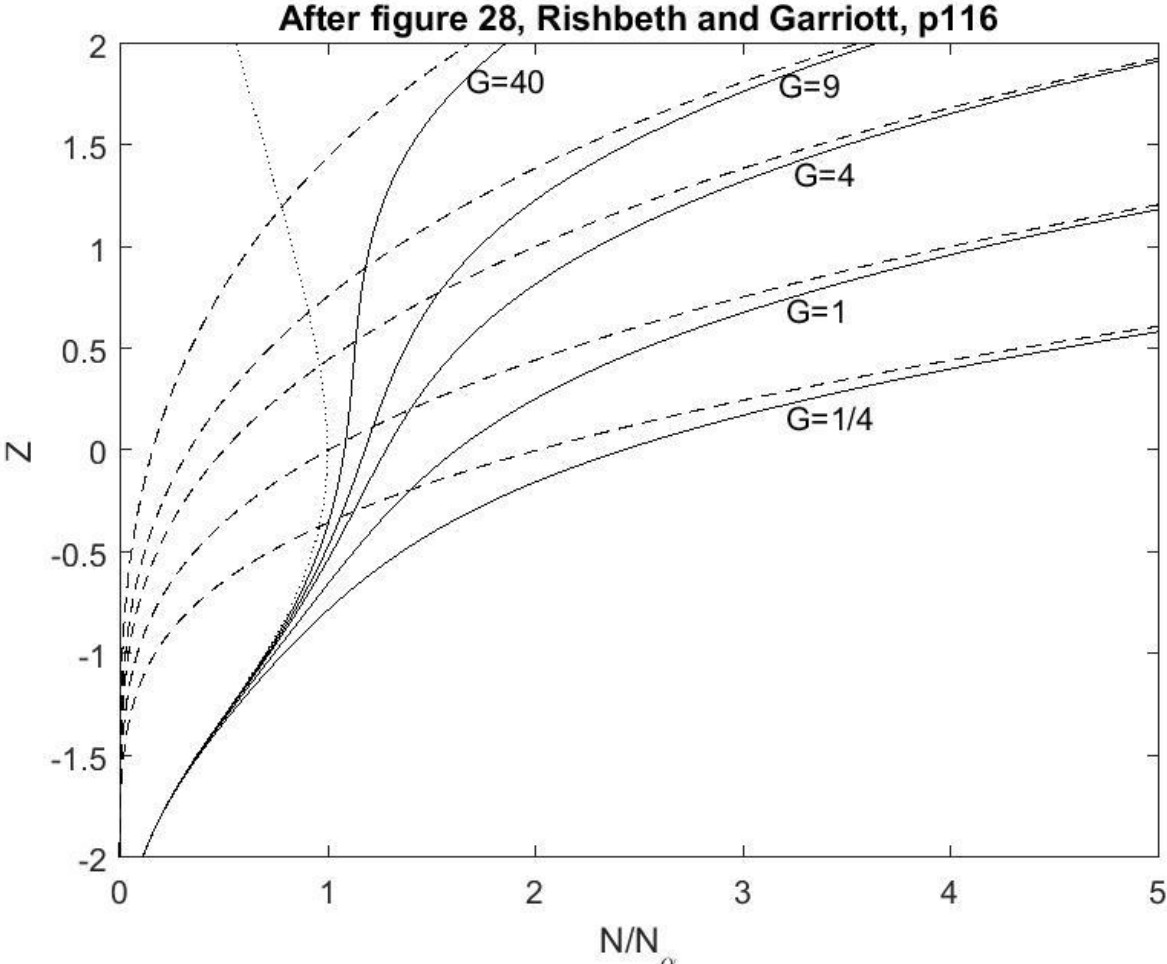

**Figure 1: Electron concentration versus reduced height for the ionospheric transition region. For a Chapman production function, q(z) with peak at z=0 and a square-law loss coefficient, α. A linear loss coefficient, $\beta=\beta_o\exp(-1.75z)$ is used, with five values of $\beta_o$ representing G values of ¼, 1, 4, 9 and 40. The dashed lines represent $N_\beta=q/\beta$. The dotted line represents $N_\alpha=(q/\alpha)^{1/2}$. The solid lines are the profiles N(z) calculated from equation (7), scaled by $N_{\alpha 0}=(q_o/\alpha)^{1/2}$. The F1 ledge is most pronounced for G=40 and absent for G=¼.**


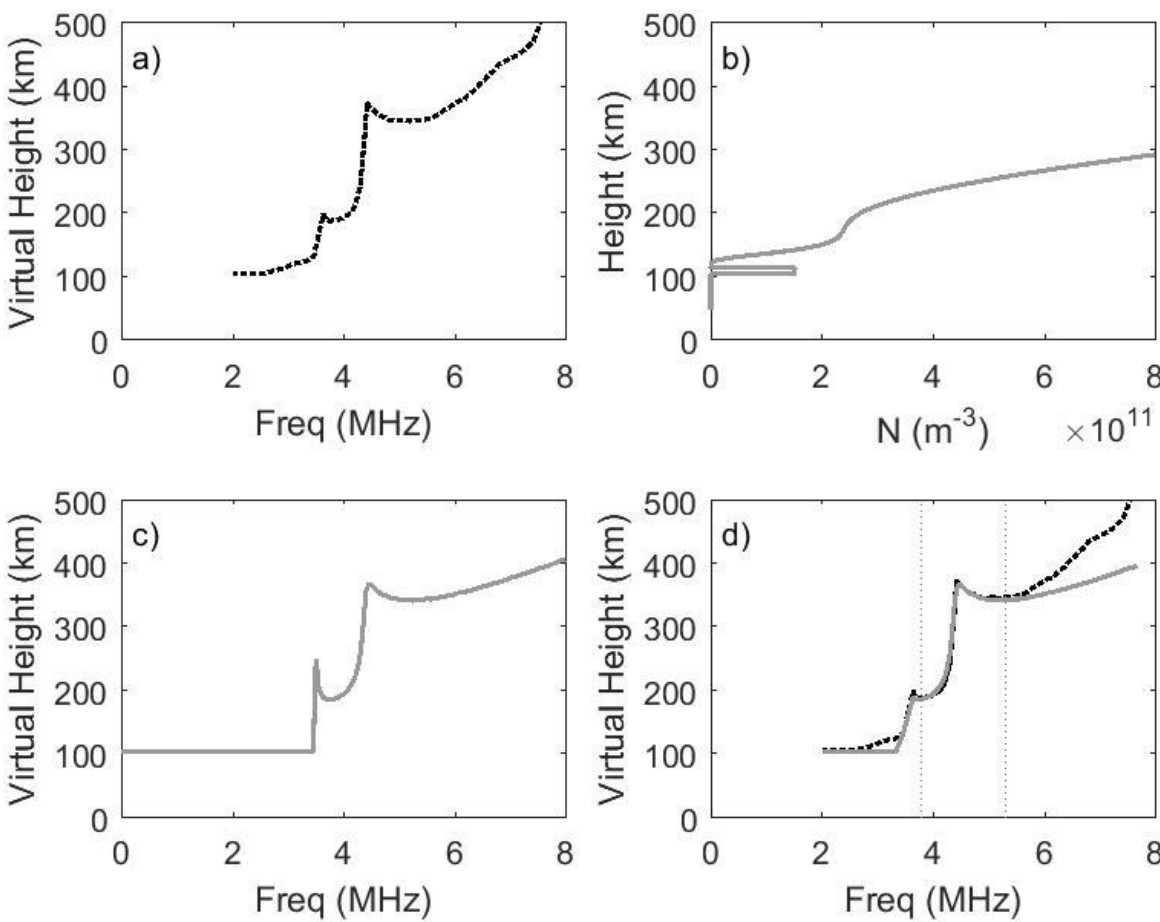


**Figure 2: An example of the fitting procedure. The virtual height profile is manually scaled from an ionogram (a). The five fit parameters are then iterated to create an electron concentration profile (b) which is in turn inverted to create an artificial ionogram (c). The values of the variables are iterated until the residual is minimised between the real and model ionograms (d). The comparison between the model ionogram (grey) and the real ionogram is only valid in the transition region (indicated by the**

**two vertical dotted lines) and so only the data between these lines are considered when calculating the minimum residual.**

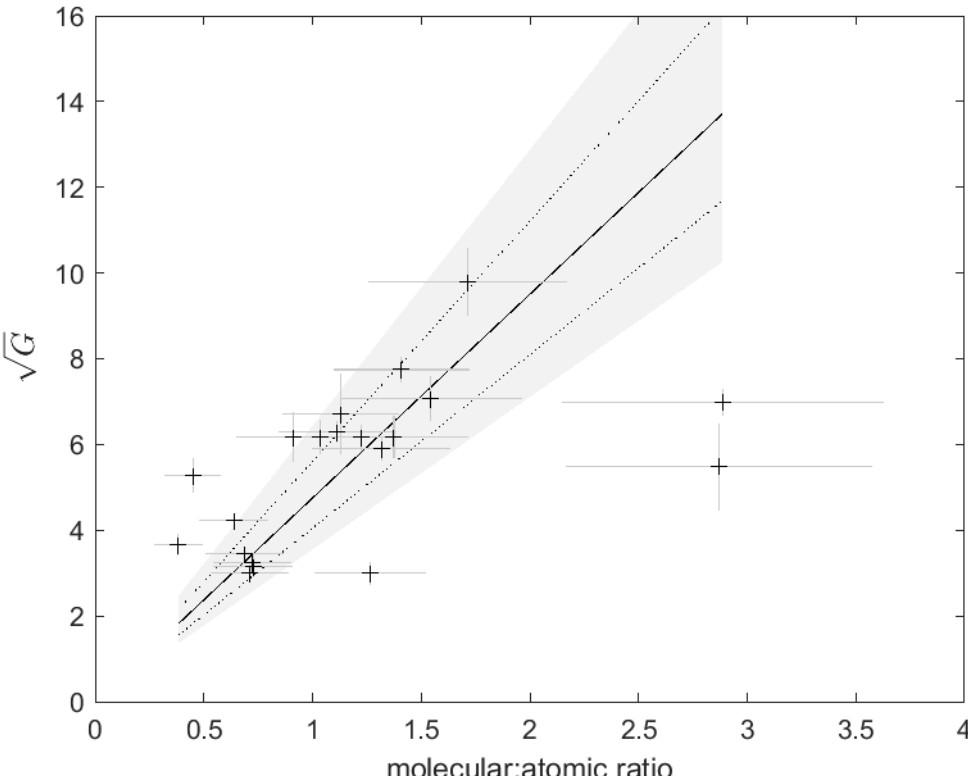

**Figure 3: A plot of √G (obtained from the model fit to ionograms) versus the thermospheric molecular:atomic ratio calculated from the associated TIMED neutral density profile. For each data point, the fitted scale height at the F1 peak was used to estimate the thermospheric temperature. This was then used to identify the appropriate height at which to interpolate the thermospheric composition ratio from the TIMED dataset. The solid line represents a weighted linear least-squares fit to the data while the dotted lines represent the 95[th] percentiles and the grey shaded area the range of gradients from a distribution of 1000 fits in which the values were randomly sampled from the uncertainty distributions. .**


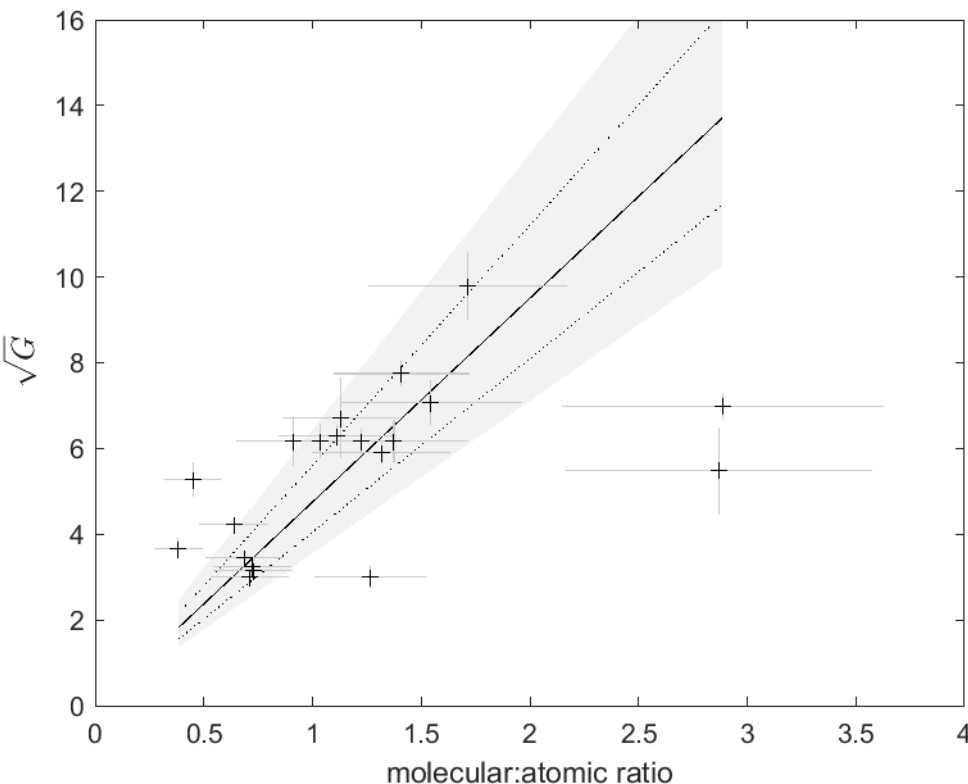

**Figure 4: The same as for figure 3 but with the heights of the F1 peak determined from the ionogram inversion algorithm ARTIST. The resulting fit yields a gradient of 4.75 ± 0.8. .**


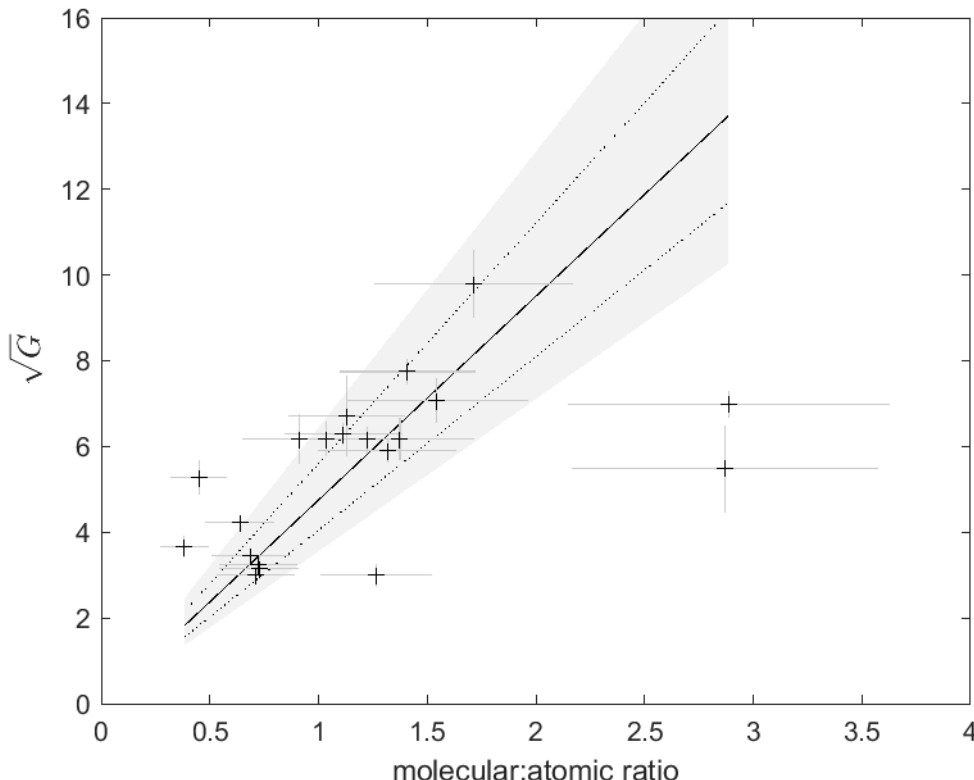

**Figure 5: The same as for figure 4 but with the analysis now extended to account for equal production in molecular and atomic species after King (1969). Once again, heights of the F1 peak were determined from the ionogram inversion algorithm ARTIST. The resulting fit yields a gradient of 4.2 ± 0.8.**

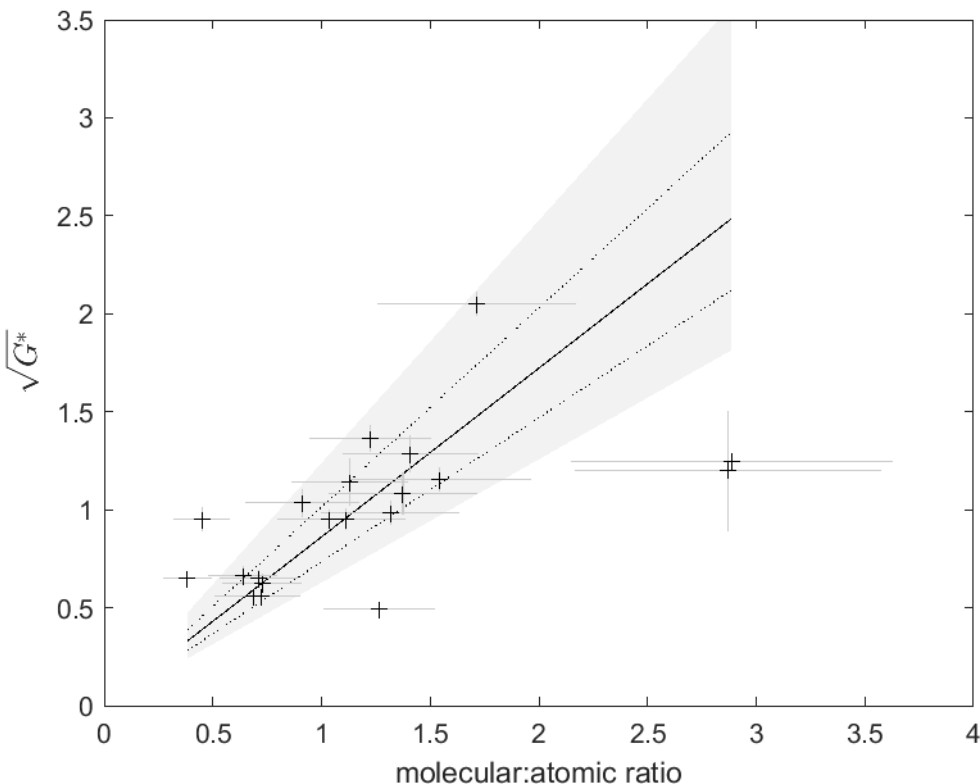


**Figure 6: The same as for figure 5 but with the alternate parameter G\* now used in the analysis, after King (1969). Once again, heights of the F1 peak were determined from the ionogram inversion algorithm ARTIST. The resulting fit yields a gradient of 0.86 ± 0.14.**

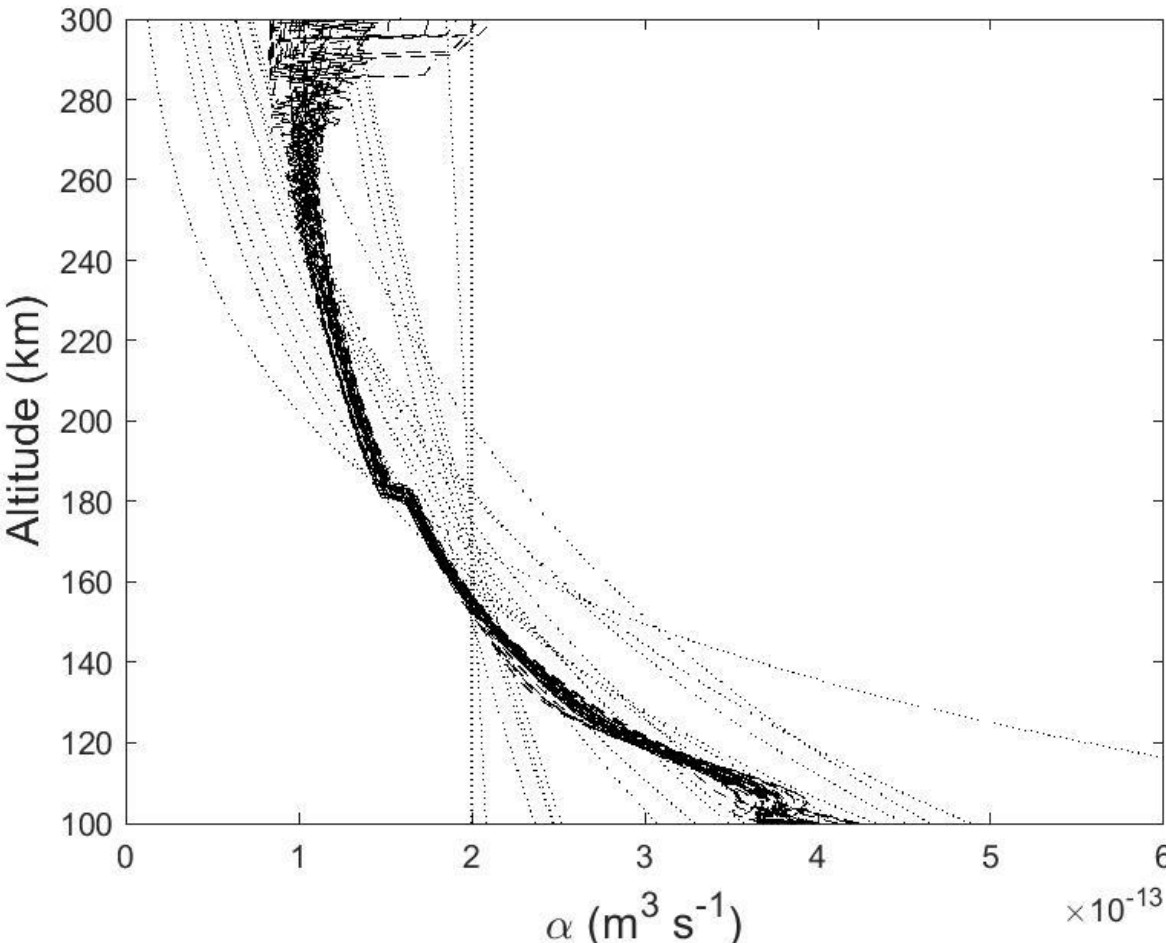


**Figure 7: A comparison between the height profiles of the molecular loss rate, α, estimated using the International Reference Ionosphere (dashed black lines) and from the simple assumption that this loss rate varies as the inverse of the scale height (dotted lines) for the overpasses used in this study.**