# Peer review of "Inferring thermospheric composition from ionogram profiles: A calibration with the TIMED spacecraft"

_Annales Geophysicae, 2019_

## Referee Comment (RC1) · Anonymous Referee #1 · 19 Apr 2019

Referee report on the paper "Inferring thermospheric composition from ionogram profiles: A calibration with the TIMED spacecraft" by Christopher J. Scott, Shannon Jones, Luke A. Barnard

An attempt is made to use very old ideas on F1-layer formation to extract any thermospheric data from ground-based ionosonde observations. I did not find in the paper any results on "thermospheric composition" retrieved from ionogram profiles which could be analyzed and compared to other observations or models.

Introduction is devoted to general description of the ionosphere and problems such as $CO_2$ impact on the upper atmosphere, long-term trends etc. which are not discussed

in the paper. The authors should formulate the problem being solved in the paper in a comparison with earlier publications in this field. Which new idea or results are they going to present in the paper. The problem is called – an inverse problem of aeronomy when aeronomic parameters (neutral composition, temperature, winds, solar EUV flux) are extracted from ionospheric observations. There are publications in this field which are not mentioned in the paper. For instance, Oliver, W. L. (1979). Incoherent scatter radar studies of the daytime middle thermosphere, Ann. Geophys., 35, 121–139. gives a method how ISR observations can be used to infer atomic oxygen and neutral temperature at F2-region height. The method by Perrone, L., & Mikhailov, A. V. (2018a). A New Method to Retrieve Thermospheric Parameters From Daytime Bottom-Side Ne(h) Observations. J. Geophys. Res. Space Physics, 123,10,200–10,212. https://doi.org/10.1029/2018JA025762 exactly solves the problem considered in the paper and this solution is more general than given in the reviewed paper.

The method used by the authors was proposed by H. Rishbeth around 50 years ago. So the authors should take this in account. The same ideas used by the authors may be found in "An introduction to the ionosphere and magnetoshere" by J.A. Ratcliffe (1972).

The majority of references are coming back to 1960-1970 as if ionospheric science has stopped since then.

The method is based on many assumptions which are not confirmed by anything.

In general the approach used in the paper does not correspond to the present day level and the paper cannot be recommended for a publication in Ann. Geophys.

Specific comments

P2 L1 While the ionosphere makes up only a small fraction ($\sim$0.01%) of the upper atmosphere In which units this 0.01% is measured?

P3 L1-5 These are well-known aspects of the thermospheric physics therefore this

paragraph may be deleted.

P4 L19 seconds) for a given ion. Not just "ion" but molecular ions NO+ and O2+ and this is important.

P4 L21 This dissociative recombination of ionisation associated with molecular gases Not molecular gases but molecular ions

P5 3 Ground-based ionospheric monitoring- the ionosonde The whole part is devoted to the history of ionospheric sounding and should be deleted from the paper.

P5 L8 Since the dominant ionisable gas is atomic oxygen This is not so, the production rates of q(N2+) and q(O2+) are comparable to q(O+) at F1-region heights.

P7 L25 All these assumptions should be confirmed.

P7 L25 that the loss rate, $\alpha$, is assumed to vary inversely with temperature and therefore H. It is known that dissociative recombination coefficients depend on Te rather than on Tn, but Te is not specified in the method.

P9 L4 It is know that F1-layer is mainly composed of molecular ions rather than O+.

---

## Referee Comment (RC2) · Anonymous Referee #2 · 27 Jun 2019

General Comments: The paper is interesting, informative, and well written. The authors present a procedure to calculate the molecular to atomic ratio from ionosonde data. While the testing of the new procedure is limited to a small number of cases, this procedure can be tested further with GOLD dayside data and then applied on the nightside where GOLD cannot measure neutral composition.

Specific comment: 20 Modelling work by Roble and Dickinson (1989), indicated that the Earth's upper atmosphere would be expected to cool by around 30-40K in response to greenhouse trapping in the lower ionosphere. I think it would be better to say the lower atmosphere instead of the lower ionosphere, in this context.

---

## Author Comment (AC1) · 16 Sep 2020

We would like to thank the referee for taking the time to review our manuscript. Below are detailed responses to the individual comments. An attempt is made to use very old ideas on F1-layer formation to extract any thermospheric data from ground-based ionosonde observations.

It was not our intention to present the work as anything other than as an investigation into the usefulness of these techniques which were developed in the 1960s. We were careful to reference the earlier work but agree that we should have posed the question in terms of the techniques that have been developed more recently. In order to do this

we have amended some opening paragraphs in the introduction to read;

"Given the potential applications for measurements of the neutral thermosphere, and the influence this has on the ionosphere, it is desirable to investigate whether the ionospheric measurements can be used to measure the thermosphere by proxy. Recent studies (Mikhailov et al, 2012; Mikhailov and Peronne, 2016; Peronne and Mikhailov, 2018) have used a sophisticated model containing comprehensive ion chemistry to generate fits to ionospheric profiles. They have shown good agreement between the neutral density derived from their model and the thermospheric density as measured by the CHAMP spacecraft (Bruinsma, S., et al, 2004). In this paper we investigate the potential of a more simplified technique developed in the 1960s (King, 1961; King and Lawden, 1964; Rishbeth and Kervin, 1968) in which the shape of the ionospheric profile measured by ground-based instrumentation is used to infer the thermospheric composition at the height of the ionospheric F1 layer peak. A comparison with co-located measurements of the thermospheric composition from the TIMED spacecraft provides an opportunity to determine if the shape of the ionogram profile in the F1 to F2 layer transition can be used to infer the thermospheric composition at the same altitudes."

I did not find in the paper any results on "thermospheric composition" retrieved from ionogram profiles which could be analyzed and compared to other observations or models.

Figure 3 presents a plot comparing the neutral compostion at the F1 peak as determined by the TIMED spacecraft with the 'G' parameter which was derived from a fit to the ionogram profiles. In this way, fits of the 'G' parameter could be used to infer composition from ionograms for which no spacecraft comparison is available.

Introduction is devoted to general description of the ionosphere and problems such as $CO_2$ impact on the upper atmosphere, long-term trends etc. which are not discussed in the paper.

We were attempting to demonstrate that there were varied applications that required an understanding of the thermospheric composition but we agree that there was more discussion than necessary. As a result we have deleted the entire paragraph beginning "Modelling work by Roble and Dickinson..."

The authors should formulate the problem being solved in the paper in a comparison with earlier publications in this field. Which new idea or results are they going to present in the paper. The problem is called – an inverse problem of aeronomy when aeronomic parameters (neutral composition, temperature, winds, solar EUV flux) are extracted from ionospheric observations. There are publications in this field which are not mentioned in the paper. For instance, Oliver, W. L. (1979). Incoherent scatter radar studies of the daytime middle thermosphere, Ann. Geophys., 35, 121–139. gives a method how ISR observations can be used to infer atomic oxygen and neutral temperature at F2-region height. The method by Perrone, L., & Mikhailov, A. V. (2018a). A New Method to Retrieve Thermospheric Parameters From Daytime Bottom-Side Ne(h) Observations. J. Geophys. Res. Space Physics, 123,10,200– 10,212. https://doi.org/10.1029/2018JA025762 exactly solves the problem considered in the paper and this solution is more general than given in the reviewed paper. The method used by the authors was proposed by H. Rishbeth around 50 years ago. So the authors should take this in account. The same ideas used by the authors may be found in "An introduction to the ionosphere and magnetoshere" by J.A. Ratcliffe (1972). The majority of references are coming back to 1960-1970 as if ionospheric science has stopped since then. In general the approach used in the paper does not correspond to the present day level and the paper cannot be recommended for a publication in Ann. Geophys.

We would like to thank the reviewer for pointing us to these references and we have gladly provided context for our study by including reference to the more recent work in our paper. These studies indeed present very comprehensive modelling to reproduce the ionospheric profiles and use these to infer thermospheric densities which are then

compared with data from the CHAMP satellite. We nonetheless feel there is also merit in investigating the relationship between the shape of ionospheric profiles and thermospheric composition using this simplified technique. If an empirical relation can be found between the two, it could be used to infer thermospheric composition from ionograms for times at which there are no direct satellite measurements of thermospheric composition.

We have amended the final paragraph of the introduction to read;

"Given the potential applications for measurements of the neutral thermosphere, and the influence this has on the ionosphere, it is desirable to investigate whether the ionospheric measurements can be used to measure the thermosphere by proxy. Recent studies (Mikhailov et al, 2012; Mikhailov and Peronne, 2016; Peronne and Mikhailov, 2018) have used a sophisticated model containing comprehensive ion chemistry to generate fits to ionospheric profiles. They have shown good agreement between the neutral density derived from their model and the thermospheric density as measured by the CHAMP spacecraft (Bruinsma, S., et al, 2004). In this paper we investigate the potential of a more simplified technique developed in the 1960s (King, 1961; King and Lawden, 1964; Rishbeth and Kervin, 1968) in which the shape of the ionospheric profile measured by ground-based instrumentation is used to infer the thermospheric composition at the height of the ionospheric F1 layer peak. A comparison with co-located measurements of the thermospheric composition from the TIMED spacecraft provides an opportunity to determine if the shape of the ionogram profile in the F1 to F2 layer transition can be used to infer the thermospheric composition at the same altitudes."

The method is based on many assumptions which are not confirmed by anything.

We agree that we should have done more to demonstrate the applicability of our assumptions, particularly that of assuming 100% O+ ions and the assumed height dependence of the molecular ion loss rate. We adopted these approaches to provide ease

of comparison with the earlier studies. We agree that assuming 100% O+ ions at the height of the F1 peak is likely to be violated often. To address this we now also present a variation on this analysis in which the true height of the F1 peak is determined from ionogram inversion (or indeed, self-consistently from the fit) and this is instead used to determine the thermospheric parameters against which the 'G' factor is compared. Independent of this result, it then becomes possible to use the fitted scale height at the F1 peak to determine a mean ion mass and thereby an estimate of the fraction of O+ ions at the F1 peak. This fraction varies from 20 to 100% O+ ions. This result, and a similar distribution of compositions obtained from the IRI for these dates justifies the referees comment.

The second major assumption, that of the molecular loss rate varying as the inverse of the scale height, was investigated by using the IRI to reconstruct this height profile for the same dates used in this study. When compared with the assumed height profiles, these demonstrated that the height variation was indeed similar in form to that suggested by the IRI but with a broad range of scatter as indicated in the attached figure. There being no systematic difference between the two suggest that the adoption of this assumption will contribute to the scatter of final results. We also determined that there was no correlation between the fitted scale-height gradient and the 'G' factor in our model, thus ensuring that an error in one quantity would not lead to a systematic error in the other in our analysis.

Specific comments P2 L1 While the ionosphere makes up only a small fraction (0.01%) of the upper atmosphere In which units this 0.01% is measured?

This is a fraction and therefore by definition is dimensionless. However the fraction is probably more in the region of 0.001% and so we have amended the value to reflect this

P3 L1-5 These are well-known aspects of the thermospheric physics therefore this paragraph may be deleted.

[Figure]

We agree that these concepts are well-known aspects of thermospheric physics, nevertheless the change in thermospheric composition with height is fundamental to the topic of the paper and we feel that retaining these lines makes the paper more self-contained for readers not as familiar with the topic as the referee.

P4 L19 seconds) for a given ion. Not just "ion" but molecular ions NO+ and O2+ and this is important.

Thank you. We have amended the sentence accordingly.

P4 L21 This dissociative recombination of ionisation associated with molecular gases Not molecular gases but molecular ions

Thank you. We have amended the sentence accordingly.

P5 3 Ground-based ionospheric monitoring- the ionosonde The whole part is devoted to the history of ionospheric sounding and should be deleted from the paper.

Since it is the visibility of the cusp at the F1 transition that is being used to draw an empirical relation with the background thermospheric composition, we feel that it is helpful to retain this short paragraph for those less familiar with the appearance of ionograms.

P5 L8 Since the dominant ionisable gas is atomic oxygen This is not so, the production rates of $q(N2+)$ and $q(O2+)$ are comparable to $q(O+)$ at F1-region heights.

Yes, we had taken on face value the results of King (1969) who extended the original technique to include production in both atomic and molecular species but we agree that this is assumption needed to be tested. We have subsequently included further analysis after King (1969) in which equal ionisation in the atomic and molecular species was considered. As presented by King (1969) we found that there was indeed very little change in the values of the G parameter estimated by extending the method in this way, though it did change the resulting fitted gradient from $4.75 \pm 0.8$ to $4.20 \pm 0.8$.

P7 L25 All these assumptions should be confirmed. While we have no independent measurement of temperature on which these reasonable assumptions are made, we have tested these assumptions further through our extended analyses (see points above and below)

P7 L25 that the loss rate, is assumed to vary inversely with temperature and therefore H. It is known that dissociative recombination coefficients depend on Te rather than on Tn, but Te is not specified in the method.

Without any independent estimate of temperature profile, it is not possible to improve easily on this approximation. However, we used the International Reference Ionosphere to reconstruct estimates of the loss rate profiles for the times and location of the ionospheric measurements used in this study. These are presented in figure 7. While the modelled values exhibit a much wider scatter than those generated by IRI, there is no systematic offset and so we conclude that this approximation will add to the scatter of points about the fitted line but will not significantly affect the final result.

P9 L4 It is know that F1-layer is mainly composed of molecular ions rather than O+ Having now also estimated the height of the F1 peak from the ARTIST inversions, this assumption has effectively been removed from our extended analysis. However, we again tested this using IRI runs for the times and location of the ionospheric data used in this study. Taking the height for the F1 peak from the ARTIST profile inversions in our analysis, we can use the satellite temperatures to estimate the mean ion mass. This in turn yielded an estimate of the ion composition. The distribution of these ion composition estimates was similar in both IRI and our analysis.

[Figure]

Figure 4: The same as for figure 3 but with the heights of the F1 peak determined from the ionogram inversion algorithm ARTIST. The resulting fit yields a gradient of 4.75 ± 0.8. .

**Fig. 1.** sqrt G vs mol-atomic ratio with heights of the F1 peak determined from the ionogram inversion algorithm ARTIST. The resulting fit yields a gradient of $4.75 \pm 0.8$.

[Figure]

Figure 5: The same as for figure 4 but with the analysis now extended to account for equal production in molecular and atomic species after King (1969). Once again, heights of the F1 peak were determined from the ionogram inversion algorithm ARTIST. The resulting fit yields a gradient of 4.2 ± 0.8.

**Fig. 2.** The same as for figure 4 but with the analysis now extended to account for equal production in molecular and atomic species after King (1969).

[Figure]

Figure 6: The same as for figure 5 but with the alternate parameter G* now used in the analysis, after King (1969). Once again, heights of the F1 peak were determined from the ionogram inversion algorithm ARTIST. The resulting fit yields a gradient of 0.86 ± 0.14.

**Fig. 3.** The same as for figure 5 but with the alternate parameter G* now used in the analysis, after King (1969). Once again, heights of the F1 peak were determined from ARTIST

Figure 7: A comparison between the height profiles of the molecular loss rate, α, estimated using the International Reference Ionosphere (dashed black lines) and from the simple assumption that this loss rate varies as the inverse of the scale height (dotted lines) for the overpasses used in this study.

**Fig. 4.** A comparison between the height profiles of the molecular loss rate, $\alpha$, estimated using the IRI and from the simple assumption that this loss rate varies as the inverse of the scale height

[Figure]

---

## Author Comment (AC2) · 16 Sep 2020

We would like to thank the referee for taking the time to review our manuscript. Below are detailed responses to the individual comments. We would indeed like to apply this technique to GOLD data and have indicated this in the paper. While we agree to the suggested wording correction when referring to Roble and Dickinson (1989) this is now a moot point since this paragraph was removed in response to a comment from referee #1.

---

## Author Response (AR3)

Response to referees

Inferring thermospheric composition from ionogram profiles: A calibration with the TIMED spacecraft by Scott et al.

The authors would like to thank the referee for their valuable comments which we address in more detail below;

*This paper revisited a technique that deduces thermospheric composition from ionospheric data, as the ionospheric profile in the transition region between F1 and F2 peaks depends on ion production rate and loss rates via ion-atom interchange reactions and dissociative recombination of molecular ions. However, their results show that a linear relationship between $\sqrt{G}$ and the molecular/atomic composition ratio, with a gradient of 2.55 ±0.40; however this gradient became to be 4.75 ± 0.4 as hmF1 is used. Supposedly, besides neutral temperature, the dynamic effect also could contribute to the formation of this transition region. The authors should discuss this issue.*

While we had mentioned the impact of atmospheric dynamics in passing in the discussion, we agree that it is important to address this earlier in the paper. We have therefore added the following paragraph;

"While the general behaviour of the F1-F2 transition is therefore controlled by thermospheric composition, transient features generated by atmospheric dynamics, such as Travelling Ionospheric Disturbances (TIDs), have long been known to temporarily affect electron concentration gradients within the ionospheric profile (e.g. Munro, 1950, Rawer, 1959). While caution should therefore be used when interpreting the shape of any individual ionospheric profile, the transient nature of a TID is revealed by viewing an ionogram in the context of the diurnal time series. In this way, the impact of TIDs can be minimised in any investigation for which a sufficiently large data set is available, though their presence would add to the scatter into any quantitative relationship between estimates of the G parameter and thermospheric composition."

And added the following references;

Munro, G. H.: Travelling disturbances in the ionosphere, Proc. Roy. Soc., A202, 208-223, 1950.

Rawer, K.; Irregularities and movements in the F-region, J. Atmos. Terr. Phys., 15, 38-42, 1959.

In addition, after discussion with the TIMED GUVI instrument team, we have also tweaked the abstract and introduction sections slightly to further emphasise how our technique potentially adds to current measurements of the thermosphere.

The abstract now begins;

"We present a method for augmenting spacecraft measurements of thermospheric composition with quantitative estimates of daytime thermospheric composition below 200 km, inferred from ionospheric data, for which there is a global network of ground-based stations. Measurements of thermospheric composition via ground-based instrumentation are challenging to make and so details about this important region of the upper atmosphere are currently sparse.."

And the introduction section now contains the following sentences;

"Far-ultraviolet remote sensing provides information on the integrated column $O/N_2$ ratio or height profiles of O and $N_2$ concentrations (via observations of the airglow profile on the limb of the Earth)."

and

"Observations of the Limb and Disk (GOLD) instrument (Eastes et al, 2017), hosted by the STS-14 commercial spacecraft, was launched into a geostationary orbit from where it makes column-integrated measurements of the thermosphere over an entire hemisphere and height profiles at the limb. Despite these advances, information about thermospheric composition is limited to dayside above around 200 km. This paper proposes a method of augmenting these spacecraft measurements with estimates of thermospheric composition below 200 km via a global network of ground-based ionospheric observatories."

We have also added the following references for the TIMED spacecraft and the GUVI measurement techniques used within our paper;

Christensen, A.B., L. J. Paxton, S. Avery, J. Craven, G. Crowley, D. C. Humm, H. Kil, R. R. Meier, C.-I. Meng, D. Morrison, B. S. Ogorzalek, P. Straus, D. J. Strickland, R., M. Swenson, R. L. Walterscheid, B. Wolven, and Y. Zhang, (2003), Initial Observations with the Global Ultraviolet Imager (GUVI) in the NASA TIMED Satellite Mission, J. Geophys. Res., 108 (A12), 1451, doi:10.1029/2003JA00991.

Christensen, A.B., R.L. Walterschied, M.N. Ross, C. Meng, L. Paxton, D. Anderson, G. Crowley, S. Avery, R. Meier, and D. Strickland, (1994), Global Ultraviolet Imager for the NASA TIMED Mission, SPIE Optical Spectroscopic Techniques and Instrumentation for Atmospheric and Space Research, 2266, 451-466.

Paxton, L. J., Schaefer, R. K., Zhang, Y., & Kil, H. (2017). Far ultraviolet instrument technology. Journal of Geophysical Research: Space Physics, 122(2), 2706-2733.

Strickland, D. J., J. S. Evans, and L. J. Paxton(1995), Satellite remote sensing of thermospheric O/N2 and solar EUV 1. Theory, J. Geophys. Res., 100, 12,217–12,226, doi:10.1029/95JA00574.

**NB in preparing the manuscript for publication, it was realised that figures 3, 4 and 5 were erroneously duplicated. We have regenerated the all the analysis figures and corrected this issue, which does not alter the quantitative values or the conclusions quoted in the text. We report this here after consultation with the topical editor.**